# Role of Dermal Factors Involved in Regulating the Melanin and Melanogenesis of Mammalian Melanocytes in Normal and Abnormal Skin

**DOI:** 10.3390/ijms25084560

**Published:** 2024-04-22

**Authors:** Tomohisa Hirobe

**Affiliations:** Department of Molecular Imaging and Theranostics, National Institute of Radiological Sciences, National Institutes for Quantum and Radiological Science and Technology, Chiba 263-8555, Japan; tmhirobe@ae.auone-net.jp; Tel.: +81-43-206-3143; Fax: +81-43-206-4138

**Keywords:** melanin, melanogenesis, melanocyte, growth factor, cytokine, epidermis, dermis, basement membrane, ultraviolet light, skin disease

## Abstract

Mammalian melanin is produced in melanocytes and accumulated in melanosomes. Melanogenesis is supported by many factors derived from the surrounding tissue environment, such as the epidermis, dermis, and subcutaneous tissue, in addition to numerous melanogenesis-related genes. The roles of these genes have been fully investigated and the molecular analysis has been performed. Moreover, the role of paracrine factors derived from epidermis has also been studied. However, the role of dermis has not been fully studied. Thus, in this review, dermis-derived factors including soluble and insoluble components were overviewed and discussed in normal and abnormal circumstances. Dermal factors play an important role in the regulation of melanogenesis in the normal and abnormal mammalian skin.

## 1. Introduction

Melanogenesis in mammalian melanocytes is controlled by their own genetic factors and by paracrine factors produced by the surrounding tissue environment. These melanocytes differentiate from neural-crest-derived melanoblasts [1] and produce special organelles named melanosomes which include brown- to black-colored eumelanin and yellow- to orange-colored pheomelanin [2]. These melanocytes are present mainly in the epidermis and hair follicles [1]. Melanosomes are transported to surrounding keratinocytes present in the epidermis and hair follicles. Eumelanin is produced from L-tyrosine and its synthesis is controlled by tyrosinase (TYR), TYR-related protein-1 (TRP-1), and dopachrome tautomerase (DCT)/TRP-2 [1,2]. On the other hand, pheomelanin is produced from cysteinyldopa which is produced via the reaction of L-tyrosine-derived dopaquinone with L-cysteine [2].

Mammalian melanogenesis is regulated by many factors derived from the surrounding tissue environment, such as the epidermis and dermis, in addition to genetic factors, namely numerous melanogenesis-related genes (coat color genes). The roles of melanogenesis-related genes and epidermal factors have been fully investigated [1,3]. However, the role of dermis has not been studied in detail. Recently, research on the role of soluble factors such as growth factors and cytokines derived from the dermis has gradually increased [4]. Additionally, studies of insoluble dermal factors such as collagen fibers and elastin fibers have been started [5]. In this review, recent studies on the regulation of mammalian melanogenesis via dermal-derived soluble and insoluble factors are overviewed and discussed in detail.

## 2. Regulation of Melanogenesis in the Normal Skin by the Tissue Environment

### 2.1. Factors Derived from the Epidermis

The mammalian epidermis and hair follicles mainly consist of keratinocytes and melanocytes [1]. Cellular interaction between keratinocytes and melanocytes is observed in the epidermis and hair follicles [1]. Keratinocytes play the most important roles in the control of melanogenesis in the epidermis and hair follicles in the several stages of skin development [6,7,8,9]. Numerous keratinocyte-derived paracrine factors (growth factors and cytokines), such as endothelin-1 (ET-1), ET-2, ET-3, stem cell factor (SCF)/steel factor/KIT ligand, hepatocyte growth factor (HGF), granulocyte-macrophage colony-stimulating factor (GMCSF), and leukemia inhibitory factor (LIF), have been reported to promote the melanogenesis of mammalian melanocytes [1,5,6,7,8,9].

### 2.2. Factors Derived from Dermal Fibroblasts

The study of melanogenic factors derived from the dermis has been delayed compared with the epidermis. Fibroblasts, the most prevalent cells in the dermis, have been studied in detail and are known to produce and release melanogenic factors toward epidermal melanocytes [4,10,11]. Paracrine factors, such as basic fibroblast growth factor (bFGF/FGF2) [12], acidic FGF (aFGF/FGF1) [12], transforming growth factor-β1 (TGF-β1) [13], keratinocyte growth factor (KGF) [14,15], SCF [14], HGF [16], interleukin (IL)-1α [13,17], IL-1β [13], Dickkopf-related protein-1 (DKK-1) [18,19], and neuregulin-1 (NRG-1) [20], have been reported to be released from dermal fibroblasts. These factors mainly control melanogenesis and skin pigmentation in mammals (Table 1).

KGF is secreted from fibroblasts and controls the proliferation of keratinocytes. KGF stimulates melanosome transfer from melanocytes to keratinocytes. The latter process is more active in light skin than in dark skin due to higher expression of the KGF receptor [21]. Moreover, KGF stimulates melanogenesis in cultured human melanocytes by increasing TYR mRNA levels [22]. KGF promotes the melanogenesis, dendritogenesis, DCT expression, and TYR activity of human melanocytes in culture [15]. The effect of KGF on melanogenesis may be elicited by the synergistic interaction with other paracrine factors such as ET-1, SCF, and HGF [15]. Moreover, IL-1α [17] stimulates melanogenesis, dendritogenesis, and TYR activity in murine melanocytes in culture.

The expression of NRG-1 is greater in dark skin than in light skin [20]. NRG-1 acts on melanocytes through its receptor v-erb-b2 avian erythroblastic leukemia viral oncogene homolog-3 (ERBB-3)/erb-b2 receptor tyrosine kinase and controls skin pigmentation. However, higher expression of ERBB-4, another receptor for NRG-1, is observed in light skin compared to dark skin. NRG-1 stimulates melanogenesis of human melanocytes in culture and a three-dimensional skin model [20].

Another factor derived from fibroblasts is pleiotrophin (PTN) [23]. The PTN receptor present in melanocytes is protein tyrosine phosphatase (PTRβ/ζ). PTN inhibits human melanogenesis by degrading the master gene of melanogenesis, microphthalmia-associated transcription factor (MITF), through ERK1/2 activation [23].

Secreted frizzled-related protein-2 (sFRP-2) also stimulates melanogenesis through upregulating MITF and TYR [24]. Cellular communication network factor-1 (CCN-1) is involved in regulating collagen synthesis [25]. CCN-1 binds with the integrin ⍺6β1 receptor and activates p38 and ERK1/2 MAPK signaling, resulting in the stimulation of melanogenesis [25].

Contrary to sFRP-2 and CCN-1, DKK-1 secreted from palmoplantar (PALM) fibroblasts greatly inhibits melanocyte differentiation and melanogenesis [19]. DKK-1 acts on the receptor complex, Frizzled and lipoprotein receptor-related protein 6 (LRP6), and inhibits MITF, DCT, and TYR [18]. However, the upregulation of Wnt inhibitory factor (WIF-1), another fibroblast-derived factor, in cultured normal human melanocytes induced the expressions of MITF and TYR, which were associated with increased melanin content andTYR activity [26]. These results suggest that the upregulation of WIF-1 stimulates the melanogenesis of normal human melanocytes.

**Table 1 ijms-25-04560-t001:** Role of dermal factors in the control of mammalian melanogenesis in normal skin.

Factors	Species	Date *	Ref.	Skin/Cell	Effect
DKK-1	Human	2004	[18]	PALM	MEL↓
DKK-1	Human	2007	[19]	PALM/3D model	MEL↓MITF/DCT/TYR↓
IL-1α	Mouse	2007	[17]	MCC	MEL↑
KGF	Human	2008	[21]	Skin	MEL↑
EP	Mouse	2008	[27]	MCC	MEL↑
KGF	Human	2010	[22]	MCC	MEL↑TYR mRNA↑
NRG-1	Human	2010	[20]	Skin	MEL↑ERBB-3↑
NRG-1	Human	2010	[20]	MCC/3D model	MEL↑
bFGF	Mouse	2011	[1]	MCC	MEL↓
KGF	Human	2013	[15]	MCC	MEL↑D/DCT/TYR↑
ET-1	Human	2013	[15]	MCC	MEL↑TYR↑
SCF	Human	2013	[15]	MCC	MEL↑TYR↑
HGF	Human	2013	[15]	MCC	MEL↑TYR↑
WIF-1	Human	2014	[26]	MCC	MEL↑MITF/TYR↑
PTN	Human	2015	[23]	MCC	MEL↓ERK1/2↑MITF↓
sFRP-2	Human	2016	[24]	Skin	MEL↑TYR↑
PDGF-BB	Human	2016	[28]	MCC	MEL↑D↑TYR↑
TGF-β1	Human	2016	[29]	Skin	MEL↓
CCN-1	Human	2018	[25]	MCC	MEL↑p38 ERK1/2↑MITF/TYR↑
EP	Human	2022	[30]	MC	EF↑MEL↑

Abbreviation: Ref., reference; MEL, melanogenesis; 3D model, reconstituted 3-dimensional skin model; MC, melanocytes; MCC, melanocytes in culture; and D, dendritogenesis; ↑, increased; ↓, decreased. Other Abbreviations are the same as Figure 1. * Published year.

### 2.3. Factors Derived from Dermal Cells Other than Fibroblasts

Contrary to the control of melanogenesis via fibroblast-derived factors, other factors derived from dermal cells, such as neutrophils, monocytes, macrophages, platelets, adipocytes, and endothelial cells, have been partially defined. One of the important factors derived from platelets is platelet-derived growth factor (PDGF) [31,32,33,34,35,36,37]. PDGF, which is present in the serum, stimulates the proliferation of mesenchymal cells [32]. PDGF can be purified from clotted blood serum, platelets, and platelet-rich plasma. PDGF is composed of four subunits, A, B, C, and D [33]. PDGF-A and -B can form homo and hetero dimers. PDGF-AA, -AB, and -BB are secreted in an active form, whereas PDGF-CC and -DD are secreted in a latent form with an N-terminal CUB domain [34]. Receptors for PDGF are composed of PDGFα and/or PDGF-β, which are dimerized on ligand-binding [35]. PDGF-AA binds to PDGFR-αα dimer only, whereas PDGF-BB binds to all receptors (PDGFR-αα, -αβ, and -ββ) [35]. PDGF stimulates the proliferation of mesoderm-derived cells, such as fibroblasts, vascular smooth muscle cells, microglia, and chondrocytes. PDGF is also a potent chemoattractant and activator of neutrophils, monocytes, and fibroblasts. PDGF is involved in regulating the synthesis and degradation of extracellular matrix (ECM) proteins and stimulates erythropoiesis and vasoconstriction [36]. Moreover, PDGF-A is released from adipocyte precursor cells, while PDGF-B is released from mature adipocytes [37]. PDGF-A is known to stimulate hair follicle regeneration [37]. By contrast, PDGF-BB supplemented to human melanocyte cultures increased melanin contents, the number of dendrites, and tyrosinase activity [28]. These results suggest that PDGF-BB is one of the dermal factors that promote melanogenesis. PDGFs may be released from platelets and act on skin cells in normal and abnormal circumstances such as bleeding, skin wounding, and skin inflammation, resulting in melanogenesis stimulation. It is possible that the frequent hyperpigmentation after skin wounding or inflammation may be due to the stimulation of melanogenesis elicited by PDGF-BB. Another source of PDGFs is vascular endothelial cells [38]. Thus, PDGFs that possess multiple origins in the dermis are involved in regulating the melanogenesis of mammalian melanocytes.

Recently, TGF-β1 derived from endothelial cells has been reported to be important for the regulation of mammalian melanogenesis. TGF-β1 inhibits the melanogenesis of human melanocytes [29]. However, the precise mechanism of the inhibition of melanogenesis via TGF-β1remains to be elucidated.

### 2.4. Factors Derived from Other Dermal Components

Another factor present in the dermis is fiber that consists of ECM. Collagen fiber is one of the major dermal components and is involved in regulating the mechanical stability of the skin [39]. EF is involved in regulating the elasticity and strength of skin in cooperation with collagen fiber [40]. A unique hexapeptide, Val-Gly-Val-Ala-Pro-Gly (VGVAPG), is repeated multiple times in human elastin molecules [40].

Recently, elastin peptides (EPs) and EF have been shown to affect the melanogenesis of mammalian melanocytes [5,27]. In mice, Chang et al. [27] reported that EP stimulated the proliferation and differentiation of mouse melanoblasts/melanocytes. In humans, EP also stimulated the melanogenesis of epidermal melanocytes in normal human skin [30]. Moreover, EFs have been reported to stimulate the melanogenesis of human epidermal melanocytes [5]. Thus, EP and EF are thought to be one of the important factors involved in mammalian melanogenesis in the skin.

What kinds of mechanism are underlined in the interaction between epidermal melanocytes and EP/EF? A study using electron microscopy revealed that melanocytes were aligned along the long axis of elastin fibers of normal dermal melanocytes of a monkey (*Cynomolgus macaques*) [41], suggesting that the direct contact of EFs with dermal melanocytes is performed to maintain the homeostasis of dermal melanocytes in the monkey. In normal embryonic mice, melanoblasts express elastin binding protein (EBP) and VGVAPG peptide stimulates murine melanogenesis [27]. The expression of EBP in the mouse melanoblasts is initiated early in skin development (E12.5) [27]. Thus, the direct interaction between dermal/epidermal melanocytes and EP/EF seems to be performed in the skin. Taken together, EF and EP are assumed to stimulate melanogenesis through the binding with EBP present in the epidermal melanocytes (Figure 1).

## 3. Regulation of Melanogenesis in UV-Exposed Skin

### Factors from Dermis-Derived Cells

Mammalian melanocytes play an important role in the protection of the body against various external stimuli such as UVs (UVA and UVB) [1,2]. UVA and UVB are the main source of solar UV. Since long wavelength UVA is capable of penetrating the dermis, dermal cells, in addition to epidermal cells, are exposed to UVA. By contrast, short wavelength UVB is capable of reaching the epidermis but it is difficult for it to penetrate the dermis. However, approximately 10% of incident UVB can reach the upper layer of the dermis. Therefore, UVB in addition to UVA can produce and release melanogenesis-stimulating factors in many kinds of cells. It has been attempted to elucidate the role of soluble and insoluble factors derived from UVA/UVB-exposed skin in human melanogenesis (Table 2).

In addition to α-melanocyte-stimulating hormone (α-MSH) [1,42], many kinds of dermal factors have been found. Nerve growth factor (NGF) acts on its receptor, NGFR, and stimulates melanogenesis and dendritogenesis [43]. IL-1α also stimulates melanogenesis in UV-exposed human skin. UVB exposures or IL-1α treatment in human skin induced a small amount of tumor necrosis factor α (TNFα) from fibroblasts, whereas combined treatment of UVB and IL-1α [44] induced 30- to 40-fold higher levels of TNFα, suggesting that UVB stimulates melanogenesis through the sequential upregulation of IL-1α and TNFα. ET-1 and SCF stimulate melanogenesis through the upregulation of TYR activity in UVB-exposed human skin [6,45]. Prostaglandin F2α (PGF_2α_) stimulates melanogenesis in UV-exposed human melanocytes in vivo and in vitro [46]. UV exposure induced overexpression of HGF [47] and sFRP-2 [24] in human skin. The stimulation of melanogenesis was elicited by the upregulation of MITF and TYR. Moreover, sFRP-2 was expressed in fibroblasts in addition to melanocytes and keratinocytes [24]. TYR activity and melanin content were increased by coculturing human melanocytes with sFRP-2-overexpressing fibroblasts. Therefore, the stimulation of melanogenesis in UV-exposed human skin may be elicited by fibroblast-derived sFRP-2 [46]. HGF increased the expression of MITF, TYR, TRP-1, and DCT in UVA/UVB-exposed human skin [47]. sFRP2 stimulates melanogenesis through an upregulation of MITF and TYR in UV-exposed human skin [48]. In UV-exposed skin, the expression of TNFα was increased [48]. Finally, EF stimulated melanogenesis in UVB-exposed vitiligo (VIT) skin [49].

The study of the interaction between epidermal melanocytes and soluble factors derived from UV-exposed skin has been performed mainly in the epidermis-derived factors. However, a recent study has shown that the dermis-derived factors also act on epidermal melanocytes in a similar fashion [1,6,7].

The stimulation of melanogenesis via UV exposures is initiated through UV-induced DNA damage in the epidermis, which triggers p53-mediated synthesis of α-MSH in human skin [42]. By contrast, in mice, α-MSH is produced and released from the intermediate lobe of the pituitary and then enters the blood stream and finally reaches the epidermis [1]. Alpha-MSH binds to the melanocortin 1 receptor (MC1R) located on the cell membrane of melanocytes. The binding of α-MSH with MC1R activates the cAMP-mediated protein kinase A (PKA) pathway, resulting in the stimulation of melanogenesis [1,6,7]. This pathway is responsible for the absorption of excess UV and causes tanning. The α-MSH-cAMP-PKA signaling pathway interacts with the protein kinase C (PKC) pathway elicited by ET-1 and the mitogen-activated protein kinase (MAPK) pathway elicited by SCF, HGF, and KGF derived from the epidermis and dermis [1,6,7]. The α-MSH-cAMP-PKA signaling pathway modulates the DNA damage response (DDR) of melanocytes [50]. Concerning DDR signaling, the upregulation of MC1R expression enhances nucleotide excision repair (NER) efficacy and genomic stability [51]. This activation promotes the repair of UV-induced DNA photoproducts, cyclobutene pyrimidine dimers (CPD), through enhancing NER. Thus, α-MSH enhances CPD removal in melanocytes after UV exposures, suggesting the important role of the α-MSH-MC1R-cAMP-PKA signaling pathway in DNA repair in the epidermal melanocytes [50]. Additionally, the MC1R genotype affects the DDR in melanocytes [52]. Moreover, the increase in DNA repair through MC1R activation may be due to the elevated levels of NER proteins, namely Xeroderma pigmentosum group C protein (XPC) and phosphorylated H2A histone family member X (γH2AX) [51]. Moreover, the stimulation of MC1R augments NER through PKA-mediated ataxia telangiectasia and Rad3-related (ATR) protein phosphorylation, which in turn stabilizes the DNA repair protein complementing XP-A (XPA) and allows its colocalization at the UV-induced DNA photolesions in the nucleus [53]. Alpha-MSH and ET-1 influence the localization of XPA in the UV-exposed melanocytes [54]. Thus, the α-MSH-MC1R-cAMP-PKA pathway and ET-1-endothelin B receptor (ETBR)-PKC pathway may regulate the DNA repair mechanism through ATR–XPA signaling [55]. Moreover, MC1R possesses multiple single nucleotide polymorphisms and affects the coat color of animals and human pigmentation in addition to the susceptibility of human skin to UV. Therefore, it should be emphasized that MC1R mutations, especially loss of function mutations, may affect the regulation of NER by α-MSH [52]. MITF is a target of the α-MSH-MC1R-cAMP-PKA pathway and MITF plays an important role in UV-induced melanogenesis and DDR signaling [56]. c-AMP-induced melanogenesis and NER seem to be two separate events, though the cAMP-induced MITF activation does not affect the NER pathway [57]. However, MED23 is reported to act as a mediator between the stimulation of melanogenesis and DNA repair by regulating the expression of MITF, and the loss of MED23 stimulated the activity of NER and decreased melanogenesis through MITF and vice versa [58]. The reason why the two reports are contradictory is not fully explained.

In a similar tendency with α-MSH, UV-induced ET-1 activation is involved in regulating melanogenesis of human melanocytes [6]. ET-1 is synthesized through p53 activation through UV-induced DNA damage. ET-1 binds to its receptor, ETBR [6]. ETBR is a G protein-coupled receptor present on the surface of melanocytes and mobilizes intracellular Ca^2+^ through the activation of the PKC pathway [59]. The binding of ET-1 with ETBR increases NER, which activates c-Jun N-terminal kinase (JNK) and p38 signaling pathways [60]. Moreover, the increase in NER through ET-1 activation may be due to the DNA damage elicited by UV exposure. The effect of α-MSH and ET-1 on ATR may mutually act as a backup in phosphorylating ATR, which additionally stimulates the DNA damage recognition and NER activity [54]. Therefore, the signaling pathways of α-MSH and ET-1 control common targets in DDR signaling Therefore, α-MSH and ET-1 play an important role in the inhibition of DNA damage and survival of melanocytes exposed to UV through activation of different receptors and signaling pathways. NGF also stimulates melanogenesis and inhibits UV-induced apoptosis through the upregulation of the antiapoptotic protein Bcl-2 [61]. NGF stimulates melanogenesis through the binding to p75 NGF receptor (NGFR).

## 4. Regulation of Melanogenesis in Abnormal Skin

### 4.1. Role of Dermis-Derived Factors in Hyperpigmentary Disorders

Examples of abnormal skin are divided into hyperpigmentation (augmented melanogenesis) and hypopigmentation (extremely reduced or null melanogenesis). Representative examples of hyperpigmentation of human skin are melasma and solar lentigines (SL).

In addition to the soluble factors derived from the dermis, direct contact of the dermal components with melanocytes may control melanogenesis in abnormal skin. One of the molecules that affect epidermal melanocytes at the dermo–epidermal junction is heparinase. The reason for hyperpigmentation in human SL may be due to heparinase-induced loss of heparan sulfate (HS) chains at the basement membrane (Table 3). Thus, HS promotes the transfer of dermis-derived factors into the epidermis [62]. Moreover, Kim et al. [63] reported an increased number and size of dermal blood vessels in the human the skin of melasma. Thus, in the skin of melasma, there was a significant relationship between vessels and melanogenesis [63]. Moreover, the increased number and size of vessels in the human skin of melasma promoted the expression of vascular endothelial growth factor (VEGF), a major angiogenic factor in the skin [63]. In the hyperpigmented skin of SL cases, irregularly branching blood vessels were observed, suggesting that the increased branching in the vessels produces and releases melanogenic factors toward epidermal melanocytes [64].

In addition to HS and VEGF, SCF in the skin of melasma [65,66]/SL [67,68] as well as in dermatofibroma (DF) [69] stimulates melanogenesis of human melanocytes. Similarly, ET-1 in SL [67], HGF in DF [69]/SL [14], KGF in SL [22,68,70]/melasma [70], IL-1α in SL [22], sFRP-2 in SL [24,71]/melasma [24], and NGF in melasma [66] stimulate melanogenesis of human melanocytes. In swine skin, KGF in SL [22] and IL-1α in SL [22] stimulates melanogenesis. However, WIF-1 inhibits human melanogenesis. Thus, the inhibition of WIF-1 expression stimulates melanogenesis and further induces melasma [72]. These results suggest that paracrine factors derived from the dermis control the hyperpigmentary disorders (Table 3).

**Table 3 ijms-25-04560-t003:** Role of dermal factors in the control of mammalian melanogenesis in abnormal skin.

Factors	Species	Date *	Ref.	Skin/Cell	Effect
SCF	Human	2001	[69]	DF	SCF↑ → MEL↑TYR↑
HGF	Human	2001	[69]	DF	HGF↑ → MEL↑TYR↑
ET-1	Human	2004	[67]	SL	ET-1↑ → MEL↑
SCF	Human	2004	[67]	SL	SCF↑ → MEL↑
SCF	Human	2006	[65]	Melasma	SCF↑ → MEL↑
VEGF	Human	2007	[63]	Melasma	VEGF↑ → MEL↑
EF	Human	2008	[73]	VLS	EF↓ → MEL↓
KGF	Human	2010	[22]	SL	KGF↑ → MEL↑
KGF	Swine	2010	[22]	SL	KGF↑ → MEL↑
IL-1α	Human	2010	[22]	SL	IL-1α↑ → MEL↑
IL-1α	Swine	2010	[22]	SL	IL-1α↑ → MEL↑
SCF	Human	2010	[68]	SL	SCF↑ → MEL↑
KGF	Human	2010	[68]	SL	KGF↑ → MEL↑
EF	Human	2010	[74]	ELT	ELT↑ → EF↓MEL↓
HS	Human	2011	[62]	SL	HS↑ → MEL↑
sFRP-2	Human	2011	[71]	SL	sFRP-2↑ → MEL↑Wnt↑
EP	Human	2012	[75]	Melanoma	EP↑ → MEL↑TYR↑
WIF-1	Human	2013	[72]	Melasma	WIF-1↓ → MEL↑
EF	Human	2013	[76]	AEGCG	MMP-2↑ → EF↓MEL↓
EF	Human	2014	[77]	ELT	ELT↑ → EF↓MEL↓
KGF	Human	2015	[70]	Melasma	KGF↑ → MEL
KGF	Human	2015	[70]	SL	KGF↑ → MEL↑
SCF	Human	2016	[66]	Melasma	SCF↑ → MEL↑
NGF	Human	2016	[66]	Melasma	NGF↑ → MEL↑
sFRP-2	Human	2016	[24]	Melasma	sFRP-2↑ → MEL↑
sFRP-2	Human	2016	[24]	SL	sFRP-2↑ → MEL↑
EF	Human	2020	[5]	VIT	EF↑ → MEL↑
HGF	Human	2021	[14]	SL	HGF↑ → MEL↑
EF	Human	2022	[78]	VIT	EF↑CSS↑ → MEL↑
EF	Human	2022	[49]	VIT	UV↑EF↑ → MELç
EP	Human	2023	[79]	VIT	EP↑ → MEL↑

Abbreviations: DF, dermatofibroma; SL, solar lentigine; VLS, vitiligoid lichen sclerosus; HS; heparan sulfate; AEGCG, annular elastolytic giant cell granuloma; MMP-2, matrix metalloproteinase-2; ELT, elastophagocytosis; CSS, culture supernatant of stem cells; ↑, increased; ↓, decreased; **→**, response; * Publication years. Other abbreviations are the same as in Figure 1 and Table 1 and Table 2.

### 4.2. Role of Dermis-Derived Factors in Hypopigmentary Disorders

VIT is one of the most common hypopigmentary disorders of the human skin characterized by achromatic or hypochromatic macules in several sites of the human skin. The incidence of VIT is 0.1–2% of humans from different races, genders, and ages. Affected epidermis is generally caused by the absence of melanin and functioning melanocytes [80,81]. Although the mechanism of VIT development and repigmentation after UVB exposures and/or skin transplantation is still unclear [79,82,83,84,85,86], melanocyte death via the infiltration of CD8^+^ T lymphocytes is one of the potent hypotheses [87]. The other hypothesis is that the melanocyte death is due to the deficiency of growth factors and cytokines released from the surrounding tissue environments [88].

Recent studies have shown that dermal fibers control the development and repigmentation of VIT skin [79]. Especially, EFs but not collagen fibers seem to be involved in regulating VIT, because EFs are dramatically decreased in VIT skin [79]. Moreover, the reduced EFs and melanocyte loss in the VIT skin can be greatly restored after UVB exposures and/or skin transplantation [79]. EFs in the repigmented skin are dramatically increased and exceed those in nonlesional skin. The EFs became thick and reached the bottom of the rete ridge and inter-rete ridge epidermis [79]. These observations suggest that EFs are involved in vitiligo development and repigmentation. Moreover, the tip of the EFs extended to the basal layer of the epidermis, suggesting that elastin molecules and EFs may control mammalian melanogenesis directly [79]. Moreover, in the regimented skin, epidermal melanoblasts/melanocytes and dermal EFs were re-differentiated [79]. The density, length, and thickness of EFs were increased via these treatments.

The hypothesis that EFs control the development and repigmentation of VIT skin seems to be supported by the assumption that the destruction of EFs induces immune infiltrate (CD8^+^ T cells) followed by VIT development and that the construction of EFs inhibits immune infiltrate followed by the repigmentation in the VIT skin, because the decrease and increase in EFs in the VIT skin correlate well with the melanocyte loss and redifferentiation, respectively. The proliferation and differentiation of melanocytes may be closely related to EF development in abnormal dermis. The infiltration of CD8^+^ T cells may be inhibited by the regeneration of EFs, whereas their infiltration may be facilitated by the degradation of EFs.

Several studies of hypopigmentary disorders other than VIT have been reported. The classical study by Ono et al. [41] revealed that in DM, where dermal melanocytes abnormally proliferate, dendrites of dermal melanocytes were aligned along the long axis of EFs. Ntayi et al. [89] reported that the melanoma cells were contacted with numerous EFs. Moreover, matrix metalloproteinase-2 (MMP-2), one of the proteases related to an invasion of tumor cells, was increased in melanoma cells cultured on the EP-coated dishes. Tian et al. [75] also reported that EPs increased TYR activity, melanin content, mRNA levels of endothelin receptor B, and c-kit level in A375 human melanoma cells in culture. Langton et al. [90] reported that the contents of EFs and epidermal melanin in the forearm of aged African-American volunteers were greatly reduced compared with the young volunteers. Similar observations have been reported in which EFs in the aged volunteers are reduced compared to the younger ones [30]. Thus, it is reasonable to assume that the disruption of EF organization is detrimental to melanocyte function [30,78,90,91].

Recently, the mechanism of the interaction between EFs and melanocytes in hypopigmentary disorders has been studied in dermal melanocytosis (DM), melanoma, aged skin, vitiligoid lichen sclerosus (VLS), annular elastolytic giant cell granuloma (AEGCG), and elastophagocytosis (ELT). The mechanism leading to EF degradation in AEGCG [76] and ELT [77] includes degradative enzymatic processes. It has been shown that in VIT, reactive oxygen species (ROS) and other free radicals (FR) are increased [76].

ROS and FR produced in the dermis may increase the expression of MMPs, which in turn reduce EFs. The reduced EFs may then induce the inflammation with granuloma formation. The inflammation then induces ELT [77] which is similar with AEGCG, developing on the VIT skin [76]. Moreover, EFs are changed in VLS, which is an autoimmune disease leading to depigmentation. This disease is associated with VIT in many cases [73]. Therefore, VLS is similar to VIT clinically, but VLS possesses the characteristics of VIT and LS histologically.

Human melanocytes are thought to be triggered by inflammation in LS in a similar tendency as VIT [73]. In LS lesions, EFs are also degraded [74]. It is possible that the loss of EFs in LS lesions is due to enzymatic digestion mediated by inflammatory cells and/or ELT, which is a kind of phagocytoses of normal and abnormal EFs by histiocytes and multinucleated giant cells [74]. The normal EFs seem to be phagocytosed by macrophages, followed by the reduction in EFs. The reduction may activate macrophages, resulting in ELT. Taken together, EF reduction in VIT and other diseases seems to be due to the degradation of EFs by the hydrolysis by enzymes such as MMP-2 and/or phagocytosis such as ELT [76].

## 5. Conclusions

Melanin and melanogenesis in mammals are regulated by many factors derived from the dermis. Regulation of melanogenesis of epidermal melanocytes may be performed by the paracrine soluble factors released from dermal cells in addition to the direct contact between epidermal melanocytes with the insoluble dermal factors such as EFs. Dermal factors play an important role in mammalian melanogenesis. Some factors stimulate melanogenesis in normal, UV-exposed, and abnormal skin. KGF and EP stimulate melanogenesis in both normal and abnormal skin. EF stimulates melanogenesis in UV-exposed and abnormal skin. Other factors affect melanogenesis in any of these contexts (Table 4). Taken together, dermal factors may affect mammalian melanogenesis regardless of the differences in the skin condition. Further investigations are required with respect to the mechanism of the signaling pathway of the dermal factors in melanocytes.

## Figures and Tables

**Figure 1 ijms-25-04560-f001:**
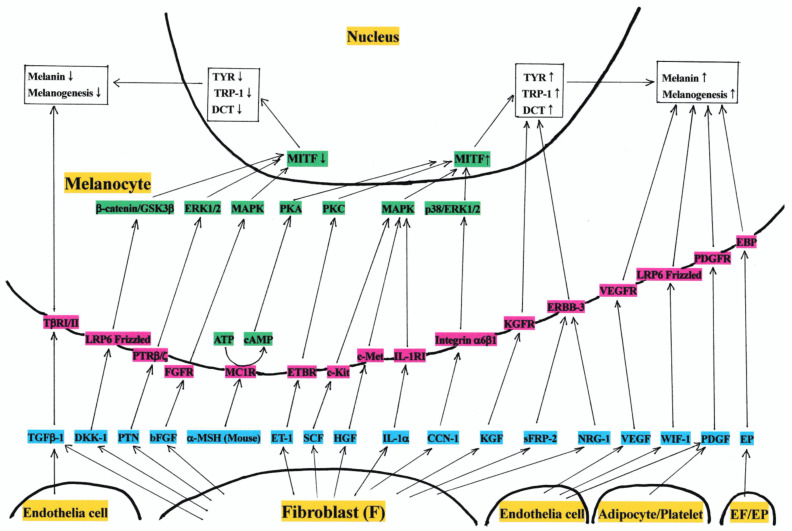
Hypothesis of the mechanism of action of dermis-derived factors on the melanogenesis of mammalian epidermal melanocytes. Abbreviations: TYR, tyrosinase; TRP-1, TYR-related protein-1; DCT, dopachrome tautomerase; MITF, microphthalmia-associated transcription factor; MAPK, MAP kinase; PKA, protein kinase A; PKC, protein kinase C; GSK3β, glycogen synthase kinase-3β; ERK 1/2, extracellular signal-regulated kinase 1/2; p38 ERK 1/2, p38 extracellular signal-regulated kinase 1/2; ATP, adenosine triphosphate; cAMP, cyclic 3′,5′ adenosine monophosphate; TβRI/II, receptor for TGFβ-1 (transforming growth factor-β1); LRP6 Frizzled, receptor for DKK-1 (Dickkopf-related protein-1) and WIF-1 (Wnt inhibitory factor-1); PTRβ/ζ, receptor for PTN (pleiotrophin)/protein tyrosine phosphatase; FGFR, receptor for bFGF (basic fibroblast growth factor); MC1R, melanocortin 1 receptor/receptor for α-MSH (melanocyte stimulating hormone); ETBR, receptor for ET-1 (endothelin-1); c-Kit, receptor for SCF (stem cell factor); c-Met, receptor for HGF (hepatocyte growth factor); IL-1RI, receptor for IL-1α (interleukin-1α); Integrin α6β1, receptor for CCN-1 (cellular communication network factor-1); KGFR, receptor for KGF (keratinocyte growth factor); ERBB-3, v-erb-b2 avian erythroblastic leukemia viral oncogene homolog/receptor for sFRP-2 (secreted frizzled-related protein-2); NRG-1, neuregulin-1; VEGFR, receptor for VEGF (vascular endothelial growth factor); PDGFR, receptor for PDGF (platelet-derived growth factor); EBP, elastin binding protein; EF, elastin fiber; EP, elastin peptide; ↑, increase; and ↓, decrease.

**Table 2 ijms-25-04560-t002:** Role of dermal factors in the control of human melanogenesis in UV-exposed skin.

Factors	Date *	Ref.	Skin/Cell	Effect
α-MSH	1995	[42]	Skin	MEL↑MITF↑TYR↑
NGF	1996	[43]	Skin	MEL↑D↑
IL-1α	1997	[44]	Skin	MEL↑
ET-1	2004	[6]	MC, MCC	MEL↑MITF↑TYR↑TRP-1↑DCT↑
SCF	2004	[6]	MC, MCC	MEL↑MITF↑TYR↑TRP-1↑DCT↑
ET-1	2004	[45]	Skin	MEL↑TYR↑
SCF	2004	[45]	Skin	MEL↑TYR↑
PGF_2α_	2005	[46]	Skin, MC, MCC	MEL↑
HGF	2010	[47]	MCC	MEL↑MITF↑TYR↑TRP-1↑DCT↑
sFRP-2	2016	[24]	Skin	MEL↑MITF↑TYR↑
TNFα	2019	[48]	Skin	MEL↑
EF	2022	[49]	VIT	MEL↑TYR↑

Abbreviation: UV, ultraviolet light; NGF, nerve growth factor; TNFα, tumor necrosis factor α; and VIT, vitiligo. Other Abbreviations are the same as in Figure 1 and Table 1. * Published year. ↑, increased; ↓, decreased.

**Table 4 ijms-25-04560-t004:** Dermal factors play an important role in the mammalian melanogenesis *.

Factors	Species	Normal	UV	Abnormal	Effect
SCF	Human	○	○	○	MEL↑TYR↑
HGF	Human	○	○	○	MEL↑TYR↑
ET-1	Human	○	○	○	MEL↑TYR↑
IL-1α	Mouse/Human/Swine	○	○	○	MEL↑
sFRP-2	Human	○	○	○	MEL↑TYR↑
NGF	Human	○	○	○	MEL↑D↑
KGF	Human/Swine	○		○	MEL↑
PDGF-BB	Human	○			MEL↑TYR↑D↑
CCN-1	Human	○			MEL↑TYR↑
NRG-1	Human	○			MEL↑
WIF-1	Human	○			MEL↑
PGF_2α_	Human		○		MEL↑
TNFα	Human		○		MEL↑
VEGF	Human			○	MEL↑
HS	Human			○	MEL↑
α-MSH	Human		○		MEL↑TYR↑MITF↑
EP	Mouse/Human	○		○	MEL↑TYR↑
EF	Human		○	○	MEL↑TYR↑
DKK-1	Human	○			MEL↓TYR↓
PTN	Human	○			MEL↓MITF↓
bFGF	Mouse	○			MEL↓
TGF-β1	Human	○			MEL↓
WIF-1	Human			○	MEL↓

* Summary of the effects of dermal factors on the mammalian melanogenesis. Normal, normal skin; Abnormal, abnormal skin; ○: the dermal factors are reported to be effective (upregulation or downregulation). Abbreviations: see the legends for Figure 1 and Table 1, Table 2 and Table 3. ↑, increased; ↓, decreased.

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
