# Peer review of "Role of Dermal Factors Involved in Regulating the Melanin and Melanogenesis of Mammalian Melanocytes in Normal and Abnormal Skin"

_ijms, 2024, doi:10.3390/ijms25084560_

Round 1

Reviewer 1 Report

Comments and Suggestions for Authors

This paper is a comprehensive review of factors (cells, ECM) in the dermis that regulate melanogenesis in normal skin, UV-irradiated skin and abnormal skin (pigment disorders). I congratulate the author on their thorough research and discussion of the literature.

The tables included for each type of skin are helpful and comprehensive.

Figure 2 is a good concept but needs to be edited as the text is too small to read properly. I suggest making this a full-page figure.

Given the paper is comparing the role of dermal-derived factors on melanogenesis in different types of skin, an overall comparative figure or table that brings this all together at the end would be useful. As it stands, each individual section is well discussed but it is hard to integrate the information. A figure or table doing this would strengthen the paper and particularly the conclusion, which at the moment is limited. 

I also think the section on hypopigmentation is quite repetitive and could be edited/streamlines - e.g. the section that starts on line 360 seems quite similar to points discussed on the previous page.

Comments on the Quality of English Language

In places, the language would be improved by some editing of linking words (such as and/the) - often "the" isn't included where it should be, and is omitted when it is required. There are numerous examples throughout the paper but these are easy/small corrections.

Some sentences could be removed without changing the meaning (e.g line 159; 230;) while others would benefit from re-wording editing (line 40, 56, 199, 210, 230, 

There are some typical graphical errors where "@" has been included instead of a symbol (eg line 60 and throughout); or a or b has been used instead of the symbol for alpha or beta (e.g. line 142, 143)

Author Response

Reviewer 1

The author totally answered to the questions and comments made by the reviewer 1 and the modified points were written in the red letter in the manuscript.

1.Figure 1 is a good concept but needs to be edited as the text is too small to read properly. I suggest making this a full-page figure.

Response: According to the comments made by the reviewer, I enlarged the Figure 1. I believe the Figure 1 becomes to be read easily and properly.

2.Given the paper is comparing the role of dermal-derived factors on melanogenesis in different types of skin, an overall comparative figure or table that brings this all together at the end would be useful. As it stands, each individual section is well discussed but it is hard to integrate the information. A figure or table doing this would strengthen the paper and particularly the conclusion, which at the moment is limited. 

Response: According to the comments made by the reviewer, I prepared and added a new Table 4. I believe the Table 4 strengthens my review article to conclude my opinion easily and properly.

  1. I also think the section on hypopigmentation is quite repetitive and could be edited/streamlines - e.g. the section that starts on line 360 seems quite similar to points discussed on the previous page.

Response: According to the comments, I totally revised the section of hypopigmentary diseases to remove the repetitive sentences.

  1. In places, the language would be improved by some editing of linking words (such as and/the) - often "the" isn't included where it should be, and is omitted when it is required. There are numerous examples throughout the paper but these are easy/small corrections.

Response: According to the comments, I totally reworded the words and sentences in all sections of my manuscript.

  1. Some sentences could be removed without changing the meaning (e.g line 159; 230;) while others would benefit from re-wording editing (line 40, 56, 199, 210, 230, 

Response: According to the suggestion, I totally removed unnecessary sentences and reworded.  

  1. There are some typical graphical errors where "@" has been included instead of a symbol (eg line 60 and throughout); or a or b has been used instead of the symbol for alpha or beta (e.g. line 142, 143)

Response: According to the suggestion, I totally modified the errors.

Reviewer 2 Report

Comments and Suggestions for Authors

The work by Dr Tomosisa Hirobe is well established on the field of melanogenesis. The current review nicely discusses the role of dermal factors including soluble and insoluble components in the regulation of melanogenesis in the normal and abnormal mammalian skin. 

1. Language editing is needed. Also, check for typos and sentence spaces.

2. In Lines 39-40: "Additionally, studies of insoluble dermal factors such as collagen fibers and elastin fibers have been started." Should a reference be added?

3. In Line 152: This paragraph starts with an important for the review question, however, it should be stated in a more emphatic, yet clear manner, connected with the rest of the 2.4 chapter. 

4. At the end of this chapter you refer to figure 1, where you form your hypothesis, without describing it al all neither there nor at the figure. Additionally, in lines 160-161: "Thus, the interaction between melanocytes and EF seems to be important for the regulation of melanogenesis (Figure 1). From the text, trying to locate this important point, it's only the last part (EF/EP) of this busy but meaningful figure. Re-organize / explain this paragraph, and the Figure if you consider it a significant point that the reader should not miss.

My opinion is that "Figure 1" should also be included in prior parts of the document (for eg. in 2.2).

5. In Lines 192-195: What did recent advances of dermal- derived paracrine factors reveal? I assume a part of the sentence is missing. This can easily happen to all of us!

Comments on the Quality of English Language

Language editing is needed. Also, check for typos and sentence spaces.

Author Response

Reviewer 2

The author totally answered to the questions and comments made by the reviewer 2 and the modified points were written in the red letters in the manuscript.

1.Language editing is needed. Also, check for types and sentence spaces.

Response: According to the suggestion, I totally modified the errors.

  1. In Lines 39-40: "Additionally, studies of insoluble dermal factors such as collagen fibers and elastin fibers have been started." Should a reference be added?

Response: According to the suggestion, I added a reference.

  1. In Line 152: This paragraph starts with an important for the review question, however, it should be stated in a more emphatic, yet clear manner, connected with the rest of the 2.4 chapter. 

Response: According to the suggestion, I totally removed the unnecessary sentences and reworded.

  1. At the end of this chapter you refer to figure 1, where you form your hypothesis, without describing it al all neither there nor at the figure. Additionally, in lines 160-161: "Thus, the interaction between melanocytes and EF seems to be important for the regulation of melanogenesis (Figure 1). From the text, trying to locate this important point, it's only the last part (EF/EP) of this busy but meaningful figure. Re-organize / explain this paragraph, and the Figure if you consider it a significant point that the reader should not miss.

Response: According to the suggestion, I totally removed the unnecessary repetitive sentences and reworded to make clear the meaning of the sentences.

5.My opinion is that "Figure 1" should also be included in prior parts of the document (for eg. in 2.2).

Response: According to the suggestion, I moved the Figure 1 before the section 2.2.

  1. In Lines 192-195: What did recent advances of dermal- derived paracrine factors reveal? I assume a part of the sentence is missing. This can easily happen to all of us!

Response: According to the comment, I totally reworded the sentences to make clear their meaning for readers.
